# Smad7 Sustains Stat3 Expression and Signaling in Colon Cancer Cells

**DOI:** 10.3390/cancers14204993

**Published:** 2022-10-12

**Authors:** Claudia Maresca, Giulia Di Maggio, Carmine Stolfi, Federica Laudisi, Marco Colella, Teresa Pacifico, Antonio Di Grazia, Davide Di Fusco, Daniele Congiu, Andrea Martina Guida, Giuseppe Sica, Ivan Monteleone, Giovanni Monteleone

**Affiliations:** 1Department of Systems Medicine, University of “Tor Vergata”, 00133 Rome, Italy; 2Department of Surgery, University of “Tor Vergata”, 00133 Rome, Italy; 3Gastroenterology Unit, Policlinico Universitario Tor Vergata, 00133 Rome, Italy

**Keywords:** colonic neoplasia, Smad, transcription factors, cytokines

## Abstract

**Simple Summary:**

Colorectal cancer (CRC) cells contain elevated levels of Stat3 and Smad7, two proteins involved in the growth and survival of neoplastic cells. This study was aimed at examining whether Smad7 positively controls the expression of Stat3 in CRC cells. By employing antisense technology, which specifically inhibits Smad7 expression in selected cells, we hereby show that the reduction of Smad7 in CRC cells is paralleled by the marked suppression of the levels of Stat3 and of Stat3-related genes. Finally, we provide evidence that in human CRC tissue samples, there is a positive correlation between Smad7 expression and Stat3 content. Overall, these findings delineate a novel positive feedback loop that sustains CRC cell behavior and suggest that Smad7 is a target for therapeutic intervention in patients with such a disease.

**Abstract:**

Colorectal cancer (CRC) cells contain elevated levels of active signal transducer and the activator of transcription (Stat)-3, which exerts proliferative and anti-apoptotic effects. Various molecules produced in the CRC tissue can activate Stat3, but the mechanisms that amplify such an activation are yet to be determined. In this paper, we assessed whether Smad7, an inhibitor of Transforiming Growth Factor (TGF)-β1 activity, sustains Stat3 expression/activation in CRC cells. Both Smad7 and phosphorylated (p)/activated-Stat3 were more expressed in the tumoral areas of CRC patients, compared to the normal adjacent colonic mucosa of the same patients, and were co-localized in primary CRC cells and CRC cell lines. The knockdown of Smad7 with a Smad7 antisense oligonucleotide (AS) reduced p-Stat3 in both unstimulated and interleukin (IL)-6- and IL-22-stimulated DLD-1 and HCT116 cells. Consistently, reduced levels of BCL-xL and survivin, two downstream signaling targets of Stat3 activation, were seen in Smad7 AS-treated cells. An analysis of the mechanisms underlying Smad7 AS-induced Stat3 inactivation revealed that Smad7 AS reduced Stat3 RNA and protein expression. A chromatin immunoprecipitation assay showed the direct regulatory effect of Smad7 on the Stat3 promoter. RNA-sequencing data from the Tumor, Normal and Metastatic (TNM) plot database showed a positive correlation between Smad7 and Stat3 in 1450 CRC samples. To our knowledge, this is the first evidence supporting the theory that Smad7 positively regulates Stat3 function in CRC.

## 1. Introduction

Colorectal cancer (CRC) is still one of the major causes of death worldwide; this is because nearly one-quarter of sufferers are diagnosed when the cancer has already metastasized. Indeed, the five-year survival rate for patients with advanced CRC is less than 10%. It is estimated that by 2030, the incidence of CRC will increase considerably, with more than 2 million new cases worldwide, resulting in more than 1 million deaths [1,2]. In the last decade, a better understanding of the factors/mechanisms that sustain cancer cell behavior has facilitated the development of effective compounds. For instance, the use of immunotherapy (e.g., immune checkpoint inhibitors targeting programmed cell death-1 or the cytotoxic T lymphocyte-associated protein-4) has contributed to promoting tumor regression in lung cancers, melanoma, and Hodgkin’s lymphoma [3]. Recent studies have shown the existence of at least two distinct CRC forms that exhibit different responses to immunotherapy. Specifically, CRCs with a high level of microsatellite instability (MSI)/deficient mismatch repair (MMR) are associated with the synthesis of cancer neoantigens and the increased presence of T cells in the tumor microenvironment. This form of CRC, which accounts for nearly 15% of stages I–III CRC and 4% of stage IV CRC, is responsive to immunotherapy [4]. In contrast, most CRCs have microsatellite-stable/proficient MMR disease and exhibit a poor response to immunotherapy [5]. This suggests the necessity of future studies aimed at identifying further druggable targets. One such target could be the signal transducer and activator of transcription (Stat)-3, as CRC cells express high levels of active Stat3 and there is evidence that such a transcription factor enhances the growth and survival of malignant cells [6,7,8]. Stat3 can be activated by various molecules that are over-produced in CRC tissue, such as interleukin (IL)-6, IL-21, IL-22, growth factors (e.g., EGFR, PDGFR), and non-receptor tyrosine kinases (e.g., Src) [9,10,11,12,13]. Nonetheless, the mechanisms that amplify Stat3 signaling in CRC are not fully understood.

Human CRC cells contain elevated levels of Smad7 [14], an inhibitor of transforming growth factor (TGF)-β1 signaling [15]. By using in vitro, ex vivo, and in vivo models of CRC, we have previously shown that the knockdown of Smad7 with a specific Smad7 antisense oligonucleotide (AS) in CRC cells promoted endoplasmic reticulum stress, thereby inhibiting cancer growth. Importantly, the anti-neoplastic action of Smad7 AS was seen in the CRC cell lines (i.e., DLD-1 and HCT116), which are not responsive to TGF-β1 [14,16], thus suggesting that the Smad7-mediated control of CRC cell growth occurs via a TGF-β-independent mechanism. In this context, it is noteworthy that recent studies have shown that Smad7 promotes the self-renewal of embryonic stem cells through a Stat3-dependent mechanism in a manner that is independent of its inhibitory effect on TGF-β1 signaling. Specifically, Smad7 directly binds to the intracellular domain of gp130 and disrupts the SHP2-gp130 or SOCS3-gp130 complex, thus expanding Stat3 signaling [17].

The aim of the present study was to assess whether Smad7 sustains the Stat3 signaling in CRC.

## 2. Materials and Methods

### 2.1. Cell Culture

All reagents were obtained from Sigma-Aldrich (Milan, Italy) unless otherwise specified. The human CRC cell lines, HCT-116 and DLD-1, were purchased from the American Type Culture Collection (ATCC, Manassas, VA, USA) and were cultured in McCoy’s 5A and RPMI 1640 medium, respectively. All media were supplemented with 10% fetal bovine serum (FBS) and 1% penicillin/streptomycin (Lonza, Verviers, Belgium).

The DLD1 and HCT116 cells were transfected with Smad7 AS or a sense oligonucleotide (2 µg/mL) for 24 h, using Opti-MEM medium and Lipofectamine 3000 reagent (Life Technologies, Milan, Italy). The phosphorothioate single-stranded oligonucleotide matching the region 107–128 (5′-GCTGCGGGGAGAAGGGGCGAC-3′) of the human Smad7 complementary DNA sequence was synthesized in both the sense and antisense orientations. DLD1 cells were also transfected with either Stat3 AS or sense oligonucleotide (both used at 100 nM) under the same conditions mentioned above. The phosphorothioate single-stranded oligonucleotide (5′-AGCTGATTCCATTGGGCCAT-3′) matching the human Stat3 complementary DNA sequence was synthesized in both the sense and antisense orientations.

To assess whether Smad7 controls cytokine-induced Stat3 phosphorylation, the above cell lines were either left untreated or transfected with Smad7 AS or sense for 24 h and finally stimulated with IL-6 (20 ng/mL, catalog number 200-06, Peprotech, London, UK) or IL-22 (20 ng/mL, catalog number 782-IL-010, R&D Systems, Minneapolis, MN, USA) for 30 min.

### 2.2. Patients and Samples

Both tumor and unaffected colonic samples were collected from 20 CRC patients who underwent surgical resection at the Tor Vergata University Hospital (Rome, Italy). Moreover, we used the TNM plot database (http://tnmplot.com/analysis/, accessed on 20 April 2022) to ascertain whether there was any correlation between Smad7 and Stat3 in human CRC.

The research was carried out according to the code of ethics of the World Medical Association (Declaration of Helsinki); informed consent was obtained, and the local ethics committee (number: 115.20) approved the study.

### 2.3. Immunofluorescence

The sections of fresh surgical specimens taken from the neoplastic areas and the normal, adjacent colonic mucosa of CRC patients and CRC cell lines were fixed with paraformaldehyde (4% final concentration), permeabilized with Triton X-100, and then blocked at room temperature with BSA 1%, Tween 0.1%, and glycine 2%. After overnight incubation at 4 °C with a rabbit primary antibody against p-Stat3 Tyr705 (1:50, catalog number 9145, Cell Signaling Technology, EuroClone, Milan, Italy) or mouse primary antibody against Smad7 (1:50, catalog number MAB2029, R&D Systems), sections were treated with a secondary antibody goat anti-rabbit Alexa488 (1:1500, catalog number A11008, Invitrogen, Waltham, MA, USA) or goat anti-mouse Alexa488 (1:1000, catalog number A11017, Invitrogen) for 1 h at room temperature. Slides were finally mounted using the ProLong gold antifade reagent with DAPI (catalog number P36931, Invitrogen), and analyzed with a LEICA DMI4000 B microscope, using LEICA application suite software (V4.6.2) (Leica, Wetzlar, Germany).

### 2.4. Western Blotting

Cells were lysed on ice in buffer containing 10 mM HEPES (pH 7.9), 0.1 mM ethylenediaminetetraacetic acid, 0.2 mM ethylene glycol-bis (β-aminoethyl ether)-N,N,N’,N’-tetraacetic acid, 10 mM potassium chloride, and 0.5% Nonidet P40, supplemented with 10 mg/mL aprotinin, 10 mg/mL leupeptin, 1 mM dithiothreitol, 1 mM phenylmethylsulphonyl fluoride, 1 mM sodium vanadate, and 1 mM sodium fluoride. The lysates were separated via sodium dodecyl sulfate-polyacrylamide gel electrophoresis. The membranes were then incubated with an antibody against Smad7 (1:1000, R&D Systems), p-Stat3 Tyr705 (1:1000, catalog number 9145, Cell Signaling Technology), total Stat3 (1:1500, catalog number 10913, Santa Cruz), BCL-xL (1:500, catalog number sc-8392, Santa Cruz), survivin (1:500, catalog number sc-17779, Santa Cruz), p-Stat1 (1:500, catalog number sc-8394, Santa Cruz), or β-actin (1:5000, catalog number A544, Sigma) and, subsequently, with a secondary antibody conjugated to horseradish peroxidase (1:20,000, rabbit anti-mouse, catalog number P0161, goat anti-rabbit, catalog number P0448, Dako, Santa Clara, CA, USA). Computer-assisted scanning densitometry was used to analyze the intensity of the immunoreactive bands.

### 2.5. Flow Cytometry

The fractions of apoptotic/necrotic cells were evaluated in cells transfected with Smad7 AS or sense, as indicated above. Cells were stained with FITC-Annexin V (AV, 1:100 final dilution, Immunotools, Friesoythe, Germany), incubated with propidium iodide (PI) (5 μg·mL^−1^) for 30 min at 4 °C, and analyzed using a Gallios flow cytometer (Beckman Coulter, Milan, Italy). In parallel experiments, cells were collected to evaluate p-Stat3 expression. Briefly, the samples were washed with PBS (2 times), fixed for 15 min at room temperature, permeabilized for 45 min at −20 °C, and then incubated with the p-Stat3 antibody, according to the manufacturer’s instructions (651004, Bio Legend).

### 2.6. Real-Time PCR

Total RNA (1 μg/sample), extracted using a PureLink mRNA mini-kit (Thermo Fisher Scientific, Waltham, MA, USA), was retrotranscribed into complementary DNA (cDNA) using Oligo(dT) primers and M-MLV-reverse transcriptase (catalog number 28025021, Thermo Fisher Scientific). This was amplified using the following conditions: denaturation for 1 min at 95 °C; annealing for 30 s at 59 °C for human Smad7 and human Stat3 and at 60 °C for human/mouse β-actin; 30 s of extension at 72 °C. RNA expression was calculated, relative to the β-actin gene, using the ΔΔCt algorithm. The primer sequences were as follows: Smad7 Fw 5′-GCCCGACTTCTTCATGGTGT-3′, Rev 5′-TGCCGCTCCTTCAGTTTCTT-3′; Stat3 Fw 5′-GGGAAGAATCACGCCTTCTA-3′, Rev 5′-ATCTGCTGCTTCTCCGTCAC-3′.

### 2.7. Chromatin Immunoprecipitation Assay

CRC cell lines were cross-linked with 1% formaldehyde for 15 min at room temperature, harvested in lysis buffer (2 × 10^7^ cells/mL), and sonicated using a Branson Sonifier 150 (20% amplitude, eight cycles consisting of 15-s pulses followed by 45-s rest periods). A chromatin immunoprecipitation (ChIP) assay was performed using Protein G magnetic beads (catalog number 1614021, BioRad, Hercules, CA, USA), in combination with the anti-Smad7 antibody (we used 5 μg, catalog number sc-365846, Santa Cruz, Dallas, TX, USA). DNA was purified using a commercially available kit (catalog number 69504, QIAGEN, Hilden, Germany) and amplified by a quantitative polymerase chain reaction (qPCR), using the following primers: human Stat3_prom Fw 5′-AAAAGGGCACAGCTGTCTCC-3′, human Stat3_prom Rev 5′-CGCTGGAGGGAAGTTTCGTT-3′, NEG Fw 5′-CATTGGGAAGTGATGATGTGATCT-3′, and NEG Rev 5′-GTCC TCTCTGCCATCTTCACTCA-3′. As a positive control, we used human c-Jun_prom Fw 5′-GGTTCTGACCAGGAGACAC, c-Jun_prom Rev 5′-CCCCTTCATGTTTCTGCTTGC-3′, according to the previously published data [18].

### 2.8. Statistical Analysis

Differences between groups were compared using Student’s *t*-test. The correlation between Smad7 and p-Stat3 in the total proteins extracted from human surgical samples and analyzed by Western blotting and between Smad7 and Stat3 RNA expression in CRC samples, as evidenced in the TNM plot database, was evaluated using the Pearson correlation coefficient and Spearman correlation, respectively. All the analyses were performed using Graph-Pad 9 software.

## 3. Results

### 3.1. In CRC, High Smad7 Is Associated with Phosphorylated Stat3 Expression

Paired surgical specimens were taken from neoplastic areas and the normal, adjacent colonic mucosa of patients with CRC and were analyzed for Smad7 and p-Stat3 via immunofluorescence. In line with previously published data [6,14], both Smad7-positive cells and p-Stat3-positive cells were more abundant in the tumor tissue, compared to the normal mucosa of CRC patients (Figure 1A,B). Double immunofluorescence showed that in CRC compartments, many cells co-expressed both Smad7 and p-Stat3 (Figure 1A,B). Consistently, Smad7 and p-Stat3 were co-localized in DLD-1 cells (Figure 1C). The specificity of the staining was confirmed by assessing Smad7 and Stat3 in DLD-1 cells transfected with Smad7 sense or AS and with Stat3 sense or AS (Appendix A).

Control isotype IgGs were used to confirm the specificity of the staining (Figure 1A, right-hand insets). Next, we assessed Smad7 and p-Stat3 in the total extracts of the tumoral samples of CRC patients via Western blotting and evaluated the relationship between these two proteins. A significant correlation between Smad7 and p-Stat3 expression was seen in the CRC samples (Figure 1D).

### 3.2. Smad7 Knockdown Reduces Stat3 Expression in CRC Cells

To test whether Smad7 controls p-Stat3, Smad7 was down-regulated in DLD-1 cells using a specific AS; 24 h later, the cell protein extracts were evaluated for p-Stat3 expression via Western blotting. The knockdown of Smad7 nearly abolished p-Stat3 expression (Figure 2A and Appendix A). The densitometry analysis of Western blot bands showed that Smad7-deficient cells exhibited significantly lower levels of p-Stat3, compared to Smad7-expressing cells (Figure 2A, right-hand panels). Similar results were seen in the HCT-116 cells (Appendix A).

When the blots were stripped and incubated with an antibody recognizing the total form of Stat3, it was evident that Smad7 knockdown was accompanied by a marked and significant down-regulation of total Stat3 (Figure 2A, Appendix A). These findings suggest that the reduced phosphorylation of Stat3 that has been seen in Smad7-deficient cells is secondary to the negative regulation of total Stat3 by Smad7 AS. In contrast, the treatment of DLD-1 cells with Smad7 AS did not affect p-Stat1 expression (Figure 2A and Appendix A), indicating that the regulatory effect of Smad7 on Stat3 expression and, hence, activation, is specific and does not reflect a general control of Smad7 on Stat molecules. These results were confirmed by flow-cytometry studies, showing that transfection of CRC cells with Smad7 AS led to a marked reduction in the percentage of p-Stat3-expressing cells (Figure 2B). Consistently, the AS-mediated inhibition of Smad7 expression was associated with the reduction of BCL-xL and survivin, two downstream signaling targets of Stat3 activation (Figure 2C, Appendix A) [19]. Moreover, at the same time point, the treatment of cells with Smad7 AS did not modify the fraction of apoptotic/necrotic cells (Figure 2D), thus excluding the possibility that the reduction of p-Stat3 in CRC cells following Smad7 knockdown is secondary to the induction of cell death.

### 3.3. Smad7 Knockdown Reduces Cytokine-Stimulated Stat3 Expression and Signaling

Recent studies in mouse embryonic stem cells have shown that Smad7 can bind to the intracellular domain of gp130 and disrupts the SHP2-gp130 or SOCS3-gp130 complex, thereby enhancing Stat3 signaling [17]. Thus, we determined whether Smad7 sustains Stat3 phosphorylation in CRC cells, perhaps through the enhancement of autocrine IL-6 signaling. To this end, we tested the effect of Smad7 AS on Stat3 phosphorylation, in response to exogenous IL-6. Immunoreactive bands corresponding to p-Stat3 expression in IL-6-stimulated DLD-1 cells were reduced using Smad7 AS treatment (Figure 3A,B and Appendix A). Densitometric analysis of the Western blot bands showed that the mean value of p-Stat3 was significantly decreased by Smad7 AS, compared to the value seen in cells treated with the sense (Figure 3). Interestingly, Smad7 knockdown also attenuated the p-Stat3 expression induced by IL-22 (Figure 3B and Appendix A), a cytokine that is over-produced in CRC tissue [20] and is able to signal in CRC cells by using a heterodimeric receptor composed of IL-22R and IL-10R2 in humans (IL-10Rβ in mice) [21,22]. Moreover, Smad7 knockdown significantly reduced the total Stat3 expression in cytokine-stimulated cells (Figure 3B and Appendix A).

### 3.4. Smad7 Knockdown Is Associated with the Down-Regulation of Stat3 RNA and Protein Expression

Recent studies have shown that Smad7 can act as a general transcription factor regulating several genes that are unrelated to the TGF-β1 pathway [23]. Therefore, we next evaluated whether the reduced Stat3 content seen in Smad7-deficient cells was secondary to the regulation of Stat3 RNA expression by Smad7. As expected, the treatment of DLD-1 with Smad7 AS significantly reduced Smad7 RNA expression in comparison to cells transfected with the sense oligonucleotide (Figure 4A). Moreover, real-time PCR showed that the Smad7-deficient cells exhibited reduced Stat3 RNA transcripts (Figure 4B).

ChIP was then performed using a specific region or different regions of the Stat3 promoter and the regulatory regions, to explore the possibility that Smad7 binds directly to sequences in the Stat3 gene. The specific region of the Stat3 promoter was chosen, based on a previous study reporting Smad7 binding to the HDAC6 promoter, through a specific consensus sequence region [18] that is also present on the promoter of Stat3. As shown in Figure 4C, Smad7 binds the Stat3 promoter in DLD-1 cells, suggesting that Smad7 directly regulates Stat3 expression.

### 3.5. Correlation between Smad7 and Stat3 in Human CRC

Finally, we used the TNM plot platform to ascertain, in human CRC samples, whether there was a correlation between Smad7 and Stat3 using the RNA-seq data. An analysis of 1450 CRC samples showed a significant positive correlation in the expression of the two genes (Figure 5).

## 4. Discussion

CRC initiation is the result of a pathological process in which both gene alterations and environmental insults trigger various intracellular signals, thereby promoting the expression/activation of several molecules involved in cell survival/growth [24]. Stat3 is involved in many cellular processes, including oncogenesis, tumor growth, and diffusion [19]. In response to cytokine and growth factor stimulation, cytoplasmic Stat3 becomes phosphorylated at tyrosine-705; this can homodimerize, as well as heterodimerize, with Stat1 and then move to the nucleus, where it initiates the transcription of several genes involved in carcinogenesis [19,25]. Consistently, Stat3 is highly activated in various types of solid tumors, including CRC [6]. These observations have boosted intensive research aimed at assessing the factors/mechanisms regulating Stat3 in CRC cells. In this study, we assessed whether, in CRC cells, Stat3 functioning can be sustained by Smad7, an intracellular protein that can regulate the expression and function of a multitude of molecules involved in tumorigenesis [26]. Our initial studies confirmed previously published findings showing that both Smad7 and p-Stat3 are up-regulated in CRC. By means of immunofluorescence, we also showed that these two proteins co-localized in CRC cells and that the knockdown of Smad7 in CRC cells selectively inhibited p-Stat3 but not Stat1. Consistently, Smad7 knockdown was accompanied by a reduced expression of the Stat3 target genes (i.e., BCL-xL and survivin). The Smad7 AS-mediated inhibitory effect on Stat3 activation was confirmed in CRC cells stimulated with Stat3-activating cytokines. Specifically, Smad7 AS reduced the p-Stat3 expression induced by IL-6 and IL-22, two cytokines that are over-produced in CRC and are able to activate Stat3 using distinct receptors [12]. Our analysis of events involved in the inactivation of Stat3 indicated that Smad7 knockdown caused a marked suppression of the Stat3 protein and RNA expression. This finding appears to be relevant, as it is known that unphosphorylated Stat3 can also undergo cytoplasmatic-nuclear shuttling and up-regulate genes that are, in part, overlapping with those up-regulated by p-Stat3 [27]. For instance, the over-expression of Stat3, which cannot be phosphorylated on residue 705, in normal human mammary epithelial cells is associated with high levels of well-known oncoproteins (e.g., MRAS and MET), as well as the genes of the Bcl-2 family [28]. Interestingly, in these cell types, Stat3 RNA transcripts can be increased by IL-6, and high levels of unphosphorylated Stat3 stimulate oncogene expression [28]. Our data are in line with such observations, as both IL-6 and IL-22 enhance the total Stat3 expression in CRC cells [12], an effect that was abolished by Smad7 knockdown. We also show that Smad7 binds to the regulatory regions of the Stat3 gene, thus providing an explanation of the reduced Stat3 expression seen in Smad7-deficient cells. However, we do not yet know whether the reduced Stat3 gene expression in response to Smad7 knockdown is either exclusively due to the direct effect of Smad7 binding to the Stat3-responsive promoter or is, in part, related to an indirect effect (e.g., the Smad7-mediated control of the molecules regulating Stat3 gene expression). The ability of Smad7 to act as a transcription factor has been previously shown in prostatic cancer cells, where Smad7 can bind the regulatory regions and enhance the expression of c-Jun and Histone deacetylase 6, two tumor-promoting genes [18]. Along the same lines is the demonstration that in mouse embryonic stem cells, Smad7 occupies the promoters of highly expressed key stemness regulators genes by binding to a specific consensus response element, NCGGAAMM [23].

The link between Smad7 and Stat3 expression in CRC was supported by means of bioinformatics analysis of a public database, showing a strong correlation between these two genes. In this context, it is noteworthy that, in addition to cancer cells, CRC samples include fractions of additional cell types (e.g., immune cells and stromal cells) that could express variable levels of Smad7 and/or Stat3 mRNA, thus contributing to the RNA transcripts measured in such samples.

Studies in human tumor cell lines have previously shown that the tumor-associated overexpression of EGFR results in the sustained hyperactivation of Stat3, which eventually induces Smad7 expression. Consistently, Smad7 mRNA and protein expression were significantly reduced in xenografts that were generated by the inoculation of Stat3-deficient tumor cells into BALB/cnu/nu mice. [29]. Although the existence of the EGFR/Stat3/Smad7 axis in CRC is yet to be evaluated, it is reasonable to hypothesize that enhanced Smad7 expression in CRC relies, at least in part, on Stat3 functioning. If this is indeed true, then it is logical to speculate that the knockdown of either Smad7 or Stat3, or perhaps the inhibition of the upstream inducers of such molecules, can interrupt a positive feedback loop that sustains CRC cell behavior.

## 5. Conclusions

To our knowledge, this is the first evidence to be published showing that Smad7 can control Stat3 expression and, hence, its activation and function in CRC cells. The present data indicate that, in addition to the well-known inhibitory function of TGF-β1 signaling, Smad7 can act as a general transcription factor, regulating several genes unrelated to the TGF-β1 pathway.

## Figures and Tables

**Figure 1 cancers-14-04993-f001:**
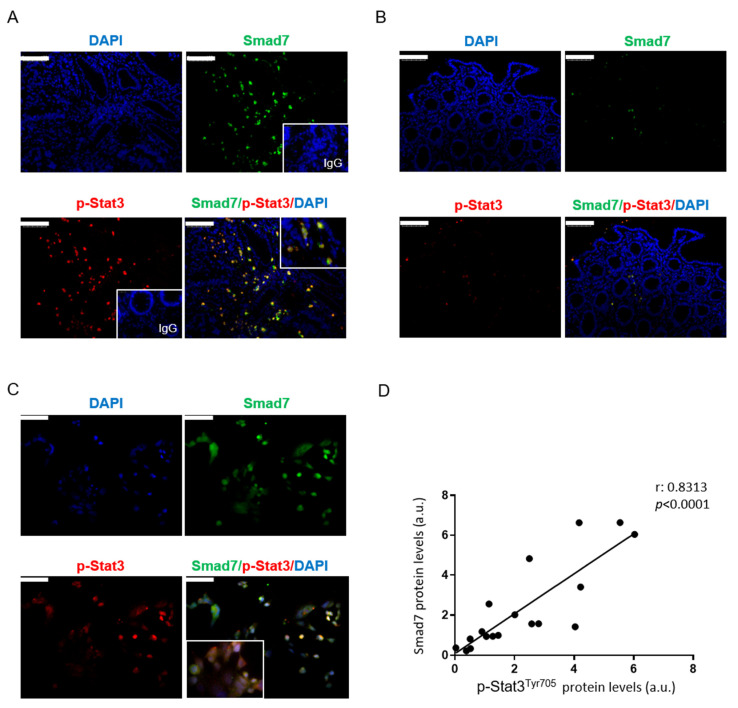
Smad7 co-localizes with p-Stat3^Y705^ and correlates with STAT3 activation in human CRC. Representative images are shown of the single- and double-immunofluorescence staining of colon sections taken from the tumoral (**A**) and non-tumoral areas (**B**) of one patient with CRC and analyzed for the expression of Smad7 (green), p-Stat3^Y705^ (red), and DAPI (blue). The scale bars are 100 μm. The insets show double-positive cells at a higher magnification and stainings with control isotype IgG. (**C**) Representative images are shown of the single- and double-immunofluorescence staining of DLD1 cells, analyzed for the expression of Smad7 (green), p-Stat3^Y705^ (red), and DAPI (blue). The scale bars are 50 μm. The figure is representative of three separate experiments in which similar results were obtained. (**D**) The correlation between the protein expression of Smad7 and p-Stat3^Y705^ in samples taken from the tumor areas of CRC patients.

**Figure 2 cancers-14-04993-f002:**
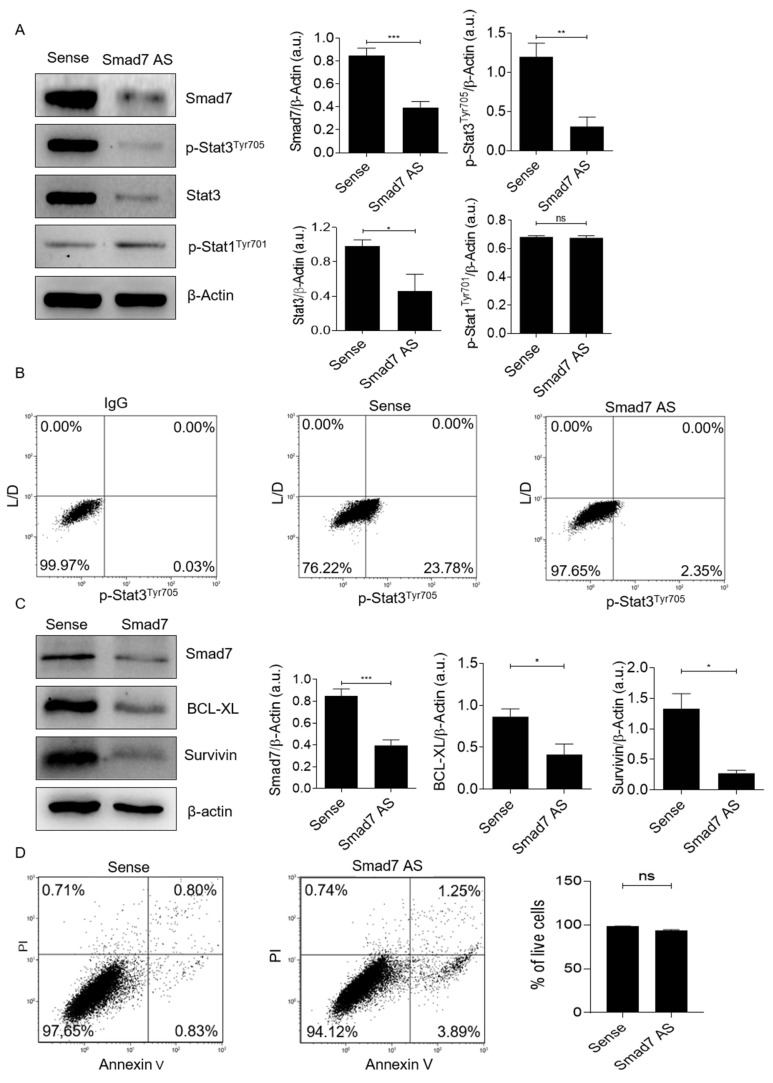
Smad7 knockdown in CRC cells reduces the expression of phosphorylated and total Stat3, as well as of Stat3-related targets. (**A**) Smad7 antisense oligonucleotide (AS) down-regulates the content of Smad7, p-Stat3^Y705^, and Stat3 in DLD1. Cells were transfected with either Smad7 AS or sense, as indicated in Section 2, then Smad7, p-Stat3, Stat3, p-Stat1^Y701^ and β-actin were analyzed via Western blotting. The right panels show a quantitative analysis of Smad7, p-Stat3, Stat3, or p-Stat1^Y701^ and β-actin, as evaluated by the densitometry scanning of Western blots. The values indicating the mean ± SEM and the differences were evaluated using a two-tailed Student’s *t*-test (* *p* < 0.05, ** *p* < 0.01, *** *p* < 0.001). (**B**) The treatment of DLD1 cells with Smad7 AS reduces the fraction of p-Stat3^Y705^-expressing cells. The cells were cultured as above and then analyzed using flow cytometry. (**C**) Smad7 AS downregulates BCL-xL and survivin protein expression. Cells were transfected, as indicated in A, and Smad7, BCL-xL, survivink and β-actin were analyzed via Western blotting. The right-hand panel shows a quantitative analysis of Smad7, BCL-xL, survivin, and β-actin, as evaluated by the densitometry scanning of Western blots. Values indicate the mean ± SEM; the differences were evaluated by a two-tailed Student’s *t*-test (* *p* < 0.05, *** *p* < 0.001). (**D**) DLD1 cells were transfected, as indicated in A, and the AV- and/or PI-positive cells were evaluated using flow cytometry. The histogram shows the fractions of AV- and/or PI-expressing cells, while the data indicate the mean ± SEM of 3 separate experiments. (ns = not statistically significant).

**Figure 3 cancers-14-04993-f003:**
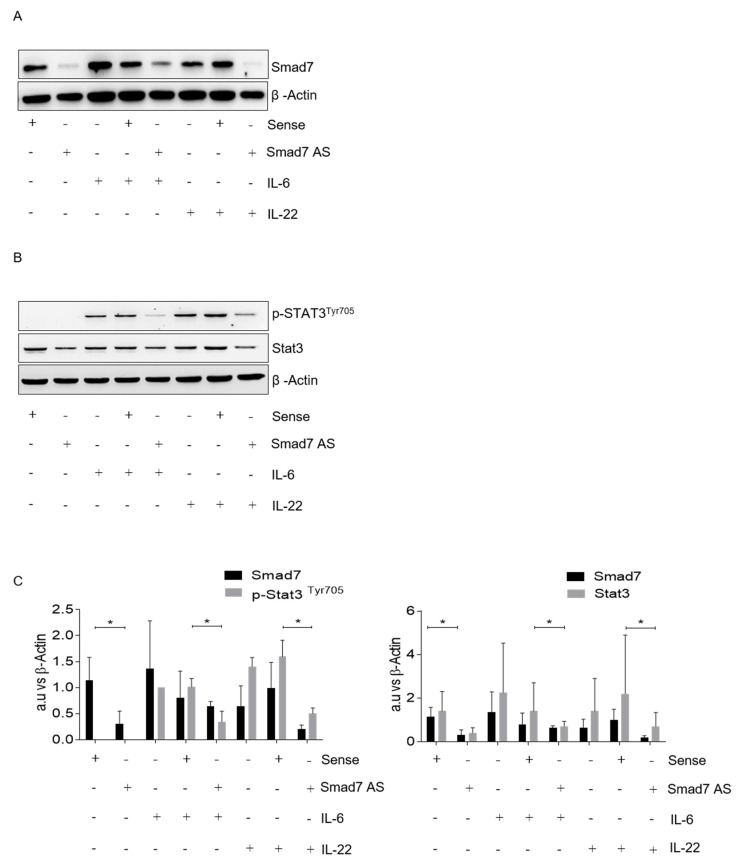
Smad7 knockdown reduces cytokine-induced p-Stat3^Y705^ and Stat3 expression in CRC cells. (**A**) DLD-1 cells were transfected with either Smad7 AS or sense for 24 h, and then stimulated or not with IL-6 or IL-22 (both used at 20 ng/mL) for 30 min. Representative Western blots for Smad7 and β-actin are shown. (**B**) Representative Western blots for p-Stat3^Y705^, Stat3, and β-actin in cells cultured as above. (**C**) Quantitative analysis of Smad7 and p-Stat3^Y705^ (left panel) and Smad7 and total Stat3 (right panel), as evaluated by the densitometry scanning of Western blots. Differences were compared using a two-tailed Student’s *t*-test (* *p* < 0.05).

**Figure 4 cancers-14-04993-f004:**
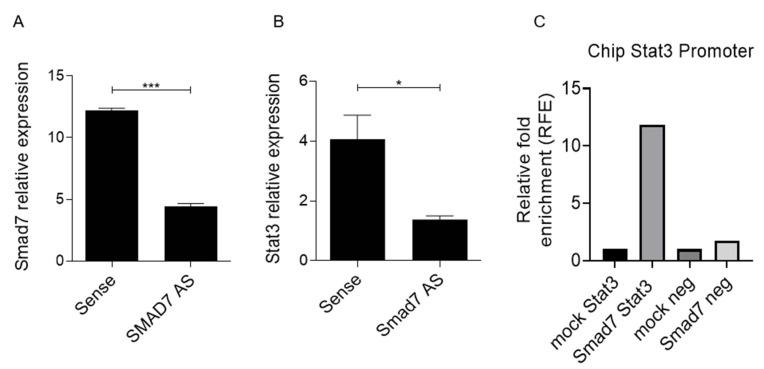
Smad7 knockdown reduces Stat3 RNA expression in CRC cells. (**A**,**B**) DLD1 cells were either transfected with Smad7 antisense (AS) or sense for 24 h. Smad7 (**A**) and Stat3 (**B**) RNA transcripts were evaluated via the real-time polymerase chain reaction. Levels are normalized to β-actin. Values show the mean ± SD of three experiments. Differences were analyzed using a two-tailed Student’s *t*-test (* *p* < 0.05, *** *p* < 0.001). (**C**) Smad7 bound the Stat3 promoter in CRC cells. Smad7 binding to the Stat3 promoters was detected via ChIP-qPCR in DLD1 cells. ChIP-qPCR data are represented as a percentage of input, normalized against a negative control sequence, compared with the binding of non-specific IgG. One out of three representative experiments is shown.

**Figure 5 cancers-14-04993-f005:**
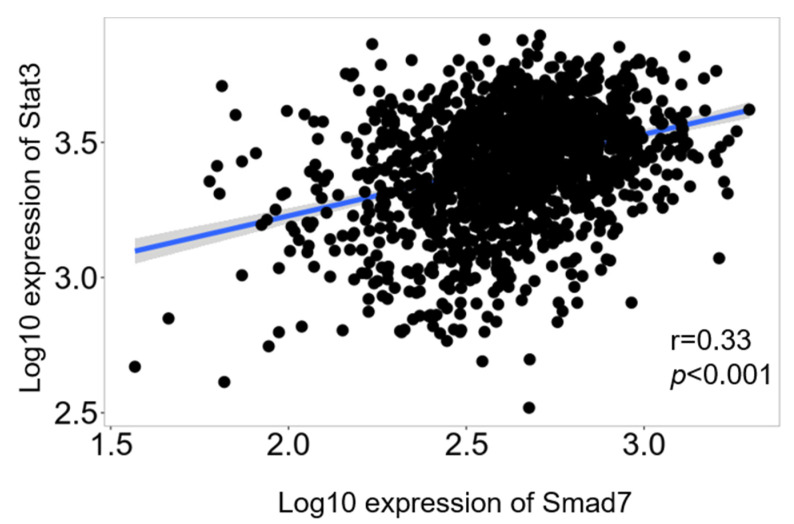
Correlation between the RNA level of Smad7 and Stat3 in human CRC. An analysis was performed on 1450 CRC samples, using the RNA-seq results present in the TNM plot database.

## Data Availability

The data presented in this study is available within the article or Appendix A.

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
