# Peer review of "Smad7 Sustains Stat3 Expression and Signaling in Colon Cancer Cells"

_cancers, 2022, doi:10.3390/cancers14204993_

Round 1

Reviewer 1 Report

The paper by Maresca et al is aims to understand the axis of SMAD7/Stat3 activation in colon cancer.

While the study per se could be of interest, it raises many concerns.

I do not think the experiments have adequate controls, and the figures do not support the conclusions of the author.

I would strongly recommend the authors to review all the figures and to avoid duplicate panels whenever possible so that the conclusions are easier to understand.

1.     I have severe concerns that Fig 4B and Fig 3 are the same western blot, but the middle line is marked with two different antibodies. Furthermore, the first two lines seems cropped out of Fig 3.

2.     A previous study by Luwor et al (Oncogene volume 32, pages 2433–2441 (2013))

 indicated the presence of a EGFR–Stat3–Smad7–TGF-β signaling axis in a different model of cancer, which is opposite of what the authors demonstrate here. Luwor and collaborator showed that knocking down Stat3 induces a downregulation of the expression on SMAD7 in vivo. The authors should add this paper to their discussion and postulate the basis of the difference with their findings.

3.     Fig1. The images are not as acceptable resolution to be able to distinguish the staining or the colocalization. The authors should provide images at a higher resolution and, for DLD1, a higher magnification may also aid the reader.  

There is no indication of a negative control being used. This is essential when stating co-localisation, as it will strengthen the findings. I would suggest the authors to provide a supplementary image with cell lines with KO of SMAD7 and cell lines with KO of Stat3, to determine the expression levels of both antibodies, as the minimal control.

Alternatively, performing in situ hybridisation with specific SMAD7 mRNA and STAT3 probe to determine concordance with the staining.

Finally, using a different antibody (recognising a different epitope), if available, could be a good option.

4.     Having Fig 2A and Fig. 4A as separate images is confusing. The SMAD7 and actin panels are the same, so it would be much easier to read if the panel with Stat3 and p-Stat3 were on the same blot. Related to this, the authors claim Smad7 induces a downregulation of activated (phospho) Stat3, but they also show evidence of downregulation of total Stat3, which is comparable in levels (just by visually evaluating the bands) with the activated form. Assuming the antibodies are specific for total and activated proteins, I do not think it is convincing to state that specifically phosphorylation is affected. The fact that the qPCR indicates a decrease in STAT3 mRNA suggest the decrease in activation is not due by SMAD7 directly, but by the decrease in amount of total Stat3 (which can be SMAD7-dependent). This may seem a little difference, but the authors should be careful in their statement when saying: “Smad7 knockdown in CRC cells reduces Stat3 phosphorylation and the expression of 200 Stat3-related targets”. Their images indicate that Smad7 knockdown reduces Stat3 expression. The authors should also perform rescue experiments, either by showing the re-expression of Stat3 once the effect of the antisense has gone, or by overexpression of Smad7 in stabile knock down cells.  

5.     Fig 4C shows the differences of Stat3 mRNA by qPCR. I would strongly recommend including the expression of SMAD7 as a control for both groups to ensure successful knock down and specificity of the antibody.

Author Response

We would like to thank the reviewer for his/her evaluation and helpful comments/suggestions.

In response to specific questions/issues raised by this reviewer:

  1. I have severe concerns that Fig 4B and Fig 3 are the same western blot, but the middle line is marked with two different antibodies. Furthermore, the first two lines seems cropped out of Fig 3.

We have taken on board the reviewer’s comment, as well as the reviewer 2’ suggestions, and  merged the original figures 2 and 3 into a new figure 2, which shows expression of both p-Stat3 and Stat3 in cells transfected with Smad7 sense or antisense oligonucleotides.

  1. A previous study by Luwor et al (Oncogene volume 32, pages 2433–2441 (2013)) indicated the presence of a EGFR–Stat3–Smad7–TGF-β signaling axis in a different model of cancer, which is opposite of what the authors demonstrate here. Luwor and collaborator showed that knocking down Stat3 induces a downregulation of the expression on SMAD7 in vivo. The authors should add this paper to their discussion and postulate the basis of the difference with their findings.

We have acknowledged and discussed the paper published by Luwor and collegaues.

  1. The images are not as acceptable resolution to be able to distinguish the staining or the colocalization. The authors should provide images at a higher resolution and, for DLD1, a higher magnification may also aid the reader.

As requested we have revised the original figure 1 and provided image at a higher resolution and magnification (DLD-1). 

  1. There is no indication of a negative control being used. This is essential when stating co-localisation, as it will strengthen the findings. I would suggest the authors to provide a supplementary image with cell lines with KO of SMAD7 and cell lines with KO of Stat3, to determine the expression levels of both antibodies, as the minimal control.

We have included images of tissue sections stained with a control antibody (revised figure 1A) as well as of cell lines with either Smad7 or Stat3 knockdown (suppl fig. 2).

  1. Having Fig 2A and Fig. 4A as separate images is confusing. The SMAD7 and actin panels are the same, so it would be much easier to read if the panel with Stat3 and p-Stat3 were on the same blot.

We have merged the two images.

b.Related to this, the authors claim Smad7 induces a downregulation of activated (phospho) Stat3, but they also show evidence of downregulation of total Stat3, which is comparable in levels (just by visually evaluating the bands) with the activated form. Assuming the antibodies are specific for total and activated proteins, I do not think it is convincing to state that specifically phosphorylation is affected. The fact that the qPCR indicates a decrease in STAT3 mRNA suggest the decrease in activation is not due by SMAD7 directly, but by the decrease in amount of total Stat3 (which can be SMAD7-dependent). This may seem a little difference, but the authors should be careful in their statement when saying: “Smad7 knockdown in CRC cells reduces Stat3 phosphorylation and the expression of 200 Stat3-related targets”. Their images indicate that Smad7 knockdown reduces Stat3 expression.

We have reworded some phrases to meet the reviewer’s suggestion

  1. Reviewer suggested us to perform rescue experiments, either by showing the re-expression of Stat3 once the effect of the antisense has gone, or by overexpression of Smad7 in stabile knock down cells.

In theory, this would be a valid suggestion but we would like to point out that our previous studies showed that transfection of CRC cells with Smad7 antisense induced a marked and persistent down-regulation of Smad7 protein expression, which was paralleled by induction of cell apoptosis. So, it is unlikely that Stat3 can be up-regulated in apoptotic cells after the antisense has gone. For the same reason, we think it is difficult to assess the effect of Smad7 over-expression in Smad7-deficient apoptotic cells.    

  1. Fig 4C shows the differences of Stat3 mRNA by qPCR. I would strongly recommend including the expression of SMAD7 as a control for both groups to ensure successful knock down and specificity of the antibody.

We have included data relative to Smad7 RNA transcripts (fig. 4A).

Reviewer 2 Report

In this article, authors demonstrate the effect of Smad7 on Stat3 activation in colon cancer cell line. This is in line with similar data reported earlier (Ref 17). Using knockdown experiments they show the reduction in Stat3 and p-Stat3 protein levels along with reduction in select down stream effector proteins (Bcl-xl, survivin etc).  They also show Smad7 colocalization pStat3 and its correlation with Stat3 expression in human colon tumors. 

Following points are needed to be addressed to improve the article...

Major:

1. Fig 1a, 1b... High resolution images of IF should be included. It is unclear which cells are positive for smad7 and pstat3. Usually these protein are abundant and widely distributed in tissue. 

2. Fig 2a... pStat1 graph should also be included. 

3. Fig 2b, 2d..... X-axis and y-axis unit markings should be included for gating

4. Fig 2c. Similar data with HCT116 is not included. Important for verifying data in another cell line.

5.Fig 3. Surprising that there is no basal pstat3 with sense+ (middle blot, 1st lane) while it was present in previous blots (Fig 2a)....Same cell line in both. Why?

6. Duplication of blots between figs. Smad7 and b-actin blots in Figs 1a and 2a are same. Smad7 and b-actin in Figs 3a and 4b are also same.  Figs 3 & 4 can be merged into one figure.

7. Smad7 and b-actin blots in Suppl Fig 1A and 1B blots are same.  Can be merged in one panel without duplication.

Minor: Several typos/grammatical errors were noted...

eg. Ln 30 "...Patients is should be patients are"

Ln 35 ...the use "of" immunotherapy....

Ln49 & 51.....Refs 7, 10, 11,12 are not cited in intro....

Discuss the status of Smad7 in CRC in the introduction.

Methods: Include cat# for items in Ln. 81, 97, 115, etc.

L 139: 2 X 10^7 cells is spelt wrong....

Ln 226...." attenuated also.."  should be "also attenuated"

Author Response

We would like to thank the reviewer for his/her evaluation and helpful comments/suggestions.

In response to specific questions/issues raised by this reviewer:

  1. Fig 1a, 1b... High resolution images of IF should be included. It is unclear which cells are positive for smad7 and pstat3. Usually these proteins are abundant and widely distributed in tissue.

We have revised the original figure 1 and provided image at a higher resolution and magnification (DLD-1). 

  1. Fig 2a... pStat1 graph should also be included.

pStat1 graph has been included

  1. Fig 2b, 2d..... X-axis and y-axis unit markings should be included for gating

The X-axis and y-axis unit markings have been included

  1. Fig 2c. Similar data with HCT116 is not included. Important for verifying data in another cell line.

Data relative to HCT116 cells have been included in the suppl fig. 1.

5.Fig 3. Surprising that there is no basal pstat3 with sense+ (middle blot, 1st lane) while it was present in previous blots (Fig 2a)....Same cell line in both. Why?

This was due to the different times of exposure of the blots. In cells stimulated with cytokines, we exposed the blots for short times in order to avoid the saturation of the bands, but this clearly made the intensity of the bands of proteins extracted from unstimulated or sense-treated cells very faint.  

6.Duplication of blots between figs. Smad7 and b-actin blots in Figs 1a and 2a are same. Smad7 and b-actin in Figs 3a and 4b are also same.  Figs 3 & 4 can be merged into one figure.

The original figures 3 and 4 have been merged into one figure (revised fig. 3).

  1. Smad7 and b-actin blots in Suppl Fig 1A and 1B blots are same. Can be merged in one panel without duplication.

The original 2 blots have been merged into one figure.

Discuss the status of Smad7 in CRC in the introduction.

The original and revised versions of the manuscript contain a wide discussion of Smad7 status in CRC.

Methods: Include cat# for items in Ln. 81, 97, 115, etc.

We included such an information

Minor: Several typos/grammatical errors were noted...

  1. Ln 30 "...Patients is should be patients are"

Ln 35 ...the use "of" immunotherapy....

L 139: 2 X 10^7 cells is spelt wrong....

Ln 226...." attenuated also.."  should be "also attenuated"

We made the requested changes.

Reviewer 3 Report

This article provides interesting and novel information to the  interplay of stat3 and smad 7 signaling in colorectal cancer. Especially through the use of the antisense nucleotide to downregulate the Smad7 transcription in CRC cells. The manuscript is very well written and have covered most of the areas. The mansucript would highly benefit in its strength if the authors could use permanent knock out models of Smad7 deficient cells to support their findings of p-stat3 regulation through Smad7 protein. Please correct some minor typos for example L50: interleukin. 

Author Response

This article provides interesting and novel information to the  interplay of stat3 and smad 7 signaling in colorectal cancer. Especially through the use of the antisense nucleotide to downregulate the Smad7 transcription in CRC cells. The manuscript is very well written and have covered most of the areas. The manuscript would highly benefit in its strength if the authors could use permanent knock out models of Smad7 deficient cells to support their findings of p-stat3 regulation through Smad7 protein.

In theory, this would be a valid suggestion but we would like to point out that our previous studies showed that knockdown of Smad7 in CRC cells induced a marked cell apoptosis. So, permanent knock-out models of Smad7 deficiency are not useful to support our findings.

Please correct some minor typos for example L50: interleukin.

We made the requested change.

Round 2

Reviewer 1 Report

The authors have addressed most of my points.

However, I still have concerns regarding figure 3A, second row (p-STAT3).

I think the authors should provide the raw images from that western blot to prove to the editor that the images have not been altered.

Author Response

Thank you for your suggestion, we have changed WB panels shown in the Fig.3. 

Round 3

Reviewer 1 Report

Thank you for choosing a better representative blot for Figure 3. However, as I think (forgive me if I am wrong in this) this blot is different from the previous one, you should also replace the lane with SMAD7 with the one belonging to the new blot. If you have not probed the new blot with SMAD7, then you should present SMAD7 lane separately with their own original actin lane. 

I apologise for being pedantic about this, but I think it is important to be transparent about western blots, which ultimately will increase the soundness of your data.

Author Response

We thank again the reviewer for his/her helpful advice. We do agree with this reviewer that the blot showing Smad7 in the revised figure 3 is similar to the previous one while we had modified the representative blots for p-Stat3, total Stat3 and b-actin. This is because we did not have access to the original, uncropped blot of Smad7. We have now taken on board the valid reviewer’s suggestion and presented the blot for Smad7 with its own original b-actin lane. In this context, we would like to point out that knockdown of Smad7 with the antisense oligonucleotide, as shown by western blotting, was confirmed in many other experiments.